# Small-Molecule Inhibitors Targeting FEN1 for Cancer Therapy

**DOI:** 10.3390/biom12071007

**Published:** 2022-07-20

**Authors:** Fan Yang, Zhigang Hu, Zhigang Guo

**Affiliations:** Jiangsu Key Laboratory for Molecular and Medical Biotechnology, College of Life Sciences, Nanjing Normal University, 1 Wenyuan Road, Nanjing 210023, China; 211202032@njnu.edu.cn

**Keywords:** FEN1, diseases, small-molecule inhibitors

## Abstract

DNA damage repair plays a key role in maintaining genomic stability and integrity. Flap endonuclease 1 (FEN1) is a core protein in the base excision repair (BER) pathway and participates in Okazaki fragment maturation during DNA replication. Several studies have implicated FEN1 in the regulation of other DNA repair pathways, including homologous recombination repair (HRR) and non-homologous end joining (NHEJ). Abnormal expression or mutation of FEN1 in cells can cause a series of pathological responses, leading to various diseases, including cancers. Moreover, overexpression of FEN1 contributes to drug resistance in several types of cancers. All this supports the hypothesis that FEN1 could be a therapeutic target for cancer treatment. Targeting FEN1 has been verified as an effective strategy in mono or combined treatment of cancer. Small-molecule compounds targeting FEN1 have also been developed and detected in cancer regression. In this review, we summarize the recent development of small-molecule inhibitors targeting FEN1 in recent years, thereby expanding their therapeutic potential and application.

## 1. Introduction

DNA is the main material basis of biological inheritance. Under normal conditions, DNA is genomically capable of maintaining a stable double helix structure. However, DNA is also vulnerable to damage by endogenous metabolites and the external environment, such as genomic instability, food containing carcinogens, and drug stimulation [1]. When experiencing endogenous or exogenous stimulation, cells can form various types of DNA damage, including base pair distortion, DNA replication errors, and the twisting and breaking of the DNA double helix [2]. The greatest risk to genomic stability is DNA double strand breaks (DSBs), and DNA damage repair pathways are crucial for maintaining genomic integrity. Any abnormal changes in the sequence of gene bases may disrupt the biochemical reactions of cells and prevent them from functioning properly [3]. DNA repair and damage signaling pathways are the key to maintaining genome stability [4]. Defects in DNA repair and damage related signaling pathways contribute to disease development, especially cancers. One of the most fundamental features of cancers is genomic instability [3]. Genomic instability can promote the development of cancer by triggering the persistent dysregulation of proto-oncogenes or tumor suppressor genes [2]. For instance, mismatch repair defects can lead to microsatellite instability, which promotes the development of colorectal and endometrial cancers [4]. Chromosomal instability is also found in most sporadic solid tumors [5]. The accumulation of DNA damage in neurons is associated with neurodegenerative diseases, including Ataxia, Alzheimer’s disease, Parkinson’s disease and other psychiatric diseases [6]. One important reason why the nervous system is susceptible to DNA damage is that the ability of adult patients’ cells to repair DNA damage is limited, which causes DNA damage to not be repaired in time, leading to accumulation, causing neuronal dysfunction and degeneration, and, finally, diseases [7]. Therefore, DNA damage and repair are of great significance for the occurrence and treatment of diseases.

Cells have different repair methods aiming at different types of DNA damage. Generally speaking, there are the following repair methods: BER, HRR, NHEJ, nucleotide excision repair (NER), mismatch repair (MMR), homology mediated double stranded DNA repair (HDR), translesion synthesis (TLS) and the SOS repair system [2]. FEN1, a structure-specific metallonuclease, is a typical member of the Rad2 nuclease family [8]. FEN1 is involved in Okazaki fragment maturation, BER, maintaining telomere stability, and the repairing of stalled DNA replication forks [9,10,11,12]. In addition, FEN1 is also involved in other major DNA metabolism pathways, including HRR, NHEJ, the disassembly of trinucleotide repeats (TNR) to sequentially derive secondary structures and the processing of apoptotic DNA fragments, etc. [13]. A large number of proteins are required to maintain genomic stability, and these proteins (e.g., PCNA) can interact with FEN1. The synergistic interaction between these proteins and FEN1 is considered to be an important molecular mechanism for FEN1 to perform related functions [14].

Tumor drug resistance is a major problem limiting the efficacy of current cancer chemotherapy drugs. The initial chemotherapy effect for many cancer patients is remarkable, but with the extension of the treatment time, cancer cells become more resistant, eventually leading to treatment failure [3]. The dysregulation of FEN1 in various tumors is closely related to tumor development and chemotherapy resistance, and is considered to be a marker of multiple tumor metastasis and poor prognosis [15]. When cisplatin was used to treat ovarian cancer, it was found that FEN1 expression was strongly induced and that the nuclear translocation of FEN1 depended on the physical interaction with Importin β. Loss; gene inactivation or inhibition of FEN1 could re-sensitize platinum-resistant ovarian cancer cells to cisplatin [16]. Ectopic expression of FEN1 in breast cancer enhances adriamycin resistance. Our previous study demonstrated that overexpression of microRNA-140 (miR-140) could enhance the drug sensitivity of breast cancer cells and reduce the drug resistance of adriamycin-resistant breast cancer cells by targeting FEN1 [17]. MiR-140 inhibited the expression of FEN1 by directly binding to its 3′ untranslated region, resulting in blocked DNA repair and thus hindering the progression of breast cancer [17]. Numerous studies have implicated FEN1 as an effective target for the treatment of tumor resistance [18]. Therefore, developing small-molecule inhibitors targeting FEN1 is a good strategy for the treatment of cancers.

## 2. Overview of FEN1

### 2.1. The Structure of FEN1

FEN1 is a 45 kDa divalent metal-dependent protein with both exonuclease and endonuclease activities, consisting of N-terminal (N), middle (I), and C-terminal (C) parts and an extended C-terminal region that can coordinate multiple DNA functions [19]. The nuclease activity of FEN1 works only in the presence of Mg^2+^ and Mn^2+^, and the presence of other salts reduces its activity [20]. FEN1 preferentially binds 5′flap by forming a hydrophobic structure around the 3′flap. This structure orients the enzyme to the bottom of the flap and allows FEN1 to cut precisely and then create a nick [21]. The active site with double Mg^2+^ atoms is located at the bottom of the flap, and the structure is sharply bent at 100°. The bend is flanked by two major protein-DNA binding sites. The 5′flap has protein binding sites only near its bottom. FEN1 first binds to the flap base, and specific amino acid-DNA interactions do not extend to the 5′flap [22]. To maintain genome integrity, FEN1 has evolved a structure that allows it to bind and bend a flap base and pass the flap through the protein prior to cleavage [23]. Figure 1 shows the spatial structures of human FEN1 (hFEN1) that are modified from the Protein Data Bank (PDB) by PyMol (https://pymol.org/2/) (accessed on 24 May 2022) [24].

### 2.2. FEN1 Functions

#### 2.2.1. Okazaki Fragment Maturation

When DNA replication occurs to a downstream Okazaki fragment, polymerase δ (Pol δ) begins to synthesize the lagging strand, terminating the 5′-end of the previous annealed Okazaki fragment to form a single-stranded 5′flap structure. FEN1 can recognize this structure, bind to the base of the flap, and cut it precisely, remove the RNA primer and some parts of the starting DNA to form the nick. To complete the maturation process, DNA ligase I (Lig I) seals the nick [25]. Thus, Okazaki fragments are thus processed and matured through the short-flap pathway. It has been demonstrated that the enzyme activity and function of FEN1 changed when point mutation occurred. When the FEN1 L209P mutant appeared in the Okazaki fragment maturation stage, the Okazaki fragment maturation will be affected and fail, resulting in the DSBs [26]. If DSBs of these DNA are not repaired or abnormally repaired, it leads to genomic instability. Cells containing the mutant had more γH2AX and 53BP1 lesions, resulting in more chromosome breaks. Further, chromosome breakage can lead to the loss of heterozygosity, activate the loss of oncogenes and tumor suppressor genes, and promote the occurrence and development of various cancers [26].

#### 2.2.2. DNA BERP Pathway

The DNA of all cells faces thousands of endogenous damages every day, which can lead to DNA instability, and BER is one of the vital ways to repair this damage [27]. There are two repair methods for BER, including short-patch BER (SP-BER), which is one nucleotide replaced and long-patch BER (LP-BER), which is more than one nucleotide replaced [28]. The successful repair of LP-BER requires three successive steps: (I) Pol β dependent polymerization to form DNA flaps, (ii) FEN1 excising the flap with its endonuclease activity, and (iii) connecting the upstream and downstream chains via DNA Lig I [29]. So, FEN1 plays an important role in the LP-BER process by cleaving the 5′flap structure [29]. FEN1 was found to interact with various proteins to exert its activity in BER. PARP (Poly ADP-ribose polymerase), a key enzyme in the repair of DNA damage, is a superfamily of multi-domain proteins [30]. Each PARP protein has a highly conserved (ADP-ribosyltransferase) domain that can catalyze the transformation of nicotinamide lysed adenine dinucleotide (NAD+) into nicotinamide and ADP-ribose [31]. During BER, FEN1 was recruited to DNA damage sites by activated PARP1 [32]. Proliferating cell nuclear antigen (PCNA) is synthesized in the nucleus and exists in the nucleus as a helper protein of DNA Pol δ, which slides onto the DNA template strand [33]. Abundant evidence indicated that hPCNA can stimulate the cleavage activity of hFEN1 and then stably bind the nuclease to its cleavage site [32]. When the FEN1 point mutation L209P is contained, the exonuclease activity of FEN1 will be interfered with, reducing the efficiency of the overall LP-BER and leading to the generation of unconnected DNA intermediates [26].

#### 2.2.3. Other DNA Repair Pathways

The N and I domains of FEN1 remain highly conserved in mammals, yeasts, fungi, bacteria, and bacteriophages. Must81 in yeast cells is an isodimer complex that performs HRR by catalyzing the decomposition of replication-related DNA structures formed during double-strand breaks (DSBs). Studies have found that FEN1 (Rad27) in yeast rapidly processes DNA damage by interacting with Must81. In addition, Rad27 plays a role in NHEJ repair pathways containing micro-homologous regions at the end of the double chain [33]. The absence of Rad27 resulted in a significant reduction in the repair efficiency of the NHEJ pathway with the end of the 5′-Flap structure, which was about 15% of that of normal strains. The repair efficiency with notches or flat ends is not affected. At the same time, Pol4 and Dnl4/Lif1 in the NHEJ pathway were shown to interact with Rad27 and stimulate its activity, so FEN1 was involved in the NHEJ repair pathway in the micro-homologous fragments [34]. Therefore, FEN1, as a key protein in DNA damage repair, effectively completes the repair of damaged DNA by interacting with the proteins of different repair pathways.

#### 2.2.4. Genome Stability

In addition to playing a role in maintaining genomic stability, FEN1 also has a phosphate diversion function that enhances 5′flap specificity and catalytic action to prevent genomic instability [35]. During Drosophila studies, Wuho (WH, GeneID: 31566; protein accession: NP_572307.1) interacted with FEN1 and acted by promoting 5′flap endonuclease activity [36]. The protein complex of WH plays a role in the protection of the genome stability at the replication fork. FEN1 can function in both direct and indirect ways in maintaining genome stability. When the homozygous FEN1 gene in mice was knocked out, the mice did not survive and the embryos were lethal [37], which proved that the importance of FEN1 to the body is unparalleled. However, after the deletion of Rad27, which is homologous to FEN1, in yeast, the maturation of Okazaki fragments promoted cell survival under stress induction [38]. The results demonstrated that the residual 5′flap could be converted into a more active 3′flap when the cells lacked the structural specific endonuclease FEN1/RAD27. When the single-stranded 3′flap formed a secondary structure, Pol δ can used this as a primer to fill in the vacant sites and connect the repeats to form atypical repeat mutants [38]. The cells used mutations as the price to survive. Therefore, the role of FEN1 in maintaining genome stability needs to be further explored.

#### 2.2.5. Telomere Stability

Telomeres consist of G-rich repeats that easily form G4 DNA. In the S and G2 phases of the cell cycle, human FEN1 is localized in telomeres and associated with the telomere repeat binding factor 2 (TRF2), a component of the spirochetes complex. FEN1 also forms an SA complex with telomerase through telomere DNA. The deficiency of FEN1 in mouse embryonic fibroblasts led to an increase in telomere-to-terminal fusion [39]. The loss of FEN1 fibroblasts leads to an increase in the γ-H2AX foci and the loss of sister telomeres copied synthetically by labeling chains [13]. The mutations that affect FEN1 gene activity and the WRN interaction sites promote telomere instability, suggesting that these two functions of FEN1 are critical to its role at the ends of chromosomes [40].

The main functions of FEN1 are illustrated in Figure 2 [11]. In addition to participating in Okazaki fragment maturation, BER, and protecting genome stability, FEN1 is also involved in multiple life processes. FEN1 is closely related to Werner syndrome protein and the telomere-binding protein interaction is required for its telomere activity [15]. Therefore, the deletion, mutation and overexpression of FEN1 in the body are bound to cause diseases.

### 2.3. Post-Transcriptional Regulation Mechanism of FEN1

The regulatory mechanism of FEN1 is mainly reflected in two aspects, substrate specificity and protein post-translational modification (PTMs) [26,41]. The FEN1 protein has a highly conserved hydrophobic pocket to recognize 3’-terminal nucleotides, which can contain single nucleotides [33]. When the active site of FEN1 was mutated (D181A), FEN1 could only bind to substrates, but not cut substrates [42]. Under normal circumstances, FEN1 performs its function during Okazaki fragment maturation. Methylation and phosphorylation are the main functions of FEN1 [43]. After methylation, FEN1 binds to PCNA and begins to remove RNA primers. When the regulatory process is completed, FEN1 will be phosphorylated and its binding with PCNA will be dissociated, providing space for the entry of DNA Lig [41]. When cells enter the late S phase, the soluble phosphorylated form of FEN1 needs to be dephosphorylated and recycled or degraded. Previous studies have demonstrated that FEN1 is degraded after the S stage, and this process is regulated by specific and sequential PTMs, including dephosphorylation, sulfonylation and ubiquitination [41]. Finally, soluble phosphorylated FEN1 is degraded by the aceylated ubiquitin proteasome pathway.

MiRNAs are a class of small non-coding RNAs with a length of about 22 nucleotides. MiRNAs are key post-transcriptional negative regulators of gene expression and are involved in the occurrence and development of various types of cancer [44]. Previous studies have demonstrated that FEN1 is regulated by a variety of miRNAs [17]. Down-regulation of MiR-140-5p is required for TGF-β-induced liver cancer metastasis, and FEN1 is a direct target of MiR-140-5p. TGF-β 1 was found to inhibit the expression of MiR-140-5p. MiR-140-5p can also inhibit the normal progress of cell cycle and inhibit the development of cervical cancer by down-regulating FEN1 [45]. MiR-140 can directly target FEN1 3′UTR and inhibit FEN1 expression. Clinical data analysis also found that miR-140 expression was down-regulated when FEN1 expression was up-regulated in breast cancer tissues. MiR-1193 could inhibit FEN1 by directly targeting YY1AP1 [46].

### 2.4. Proteins Interacting with FEN1

Protein–protein interactions are critical in guiding FEN1 into different biochemical metabolic pathways. FEN1 interacts with different proteins to form specific protein complexes that play specific roles in DNA replication and repair. These proteins include PCNA, replication protein A (RPA), apurinic/apyrimidinic endonuclease 1 (APE1), PARP1, polymerases, DNA Replication helicase/nuclease 2 (Dna2), and P300, etc. [41]. Figure 3 shows the interaction between FEN1 and different proteins [10,16]. When interacting with PCNA, FEN1 will be recruited to the replication defect region to excise the RNA primers and repair the DNA BER sites. In addition, PCNA can also strongly stimulate the endonuclease and exonuclease activities of FEN1 [45]. At the same time, in response to the stalled replication bifurcation, WRN can form a complex with FEN1 to activate its gap-dependent endonuclease activity, thus initiating the recombination repair of the broken chain [29]. It has been demonstrate that Dna2 can form a complex with FEN1 in vivo, and RPA mediated the nuclease switch between FEN1 and Dna2 [47]. RPA can inhibit FEN1 from cutting DNA while promoting the nuclease activity of Dna2. This combination reduces the in vitro processing of Okazaki fragments, thus forming the ‘dual nuclease’ Okazaki fragment processing model [47]. In addition to its role in DNA replication and repair, FEN1 degrades apoptotic genomic DNA fragments by interacting with endonuclease during apoptosis [43]. P300 is a typical acetylase and has regulatory effects on a variety of DNA replication and repair proteins. Studies have demonstrated that P300 inhibited the activity of FEN1 by acetylation. P300, on the other hand, acetylates Dna2, stimulating its 5′→3′ endonuclease, 5′→3′ helicase and DNA-dependent ATPase activities. The stability of DNA can be enhanced by changing the ratio of short and long Okazaki segments through the sequence of FEN1 and Dna2 processing of Okazaki segments [48]. Studies have found that FEN1 inhibited extensive new chain degradation of MRE11 at the bifurcation of stalled replication by regulating BRCA1-RAD51 and WRN hydrolase [49]. In clinical treatment, gene deletion or molecular inhibition of FEN1 and DNA PKC interfered with glioma progression.

## 3. The Role of FEN1 in Diseases

### 3.1. Cancers

FEN1 has been considered to be a tumor suppressor [50]. Its expression is frequently altered by disharmony or mutation in cancer, and these changes contribute to genomic instability and cancer progression [11]. FEN1 is highly expressed in a variety of tumors, including prostate cancer, neuroblastoma, pancreatic and breast cancer, lung cancer, testicular cancer, glioblastoma, and astrocytoma and so on [51,52,53,54,55]. The expression of FEN1 in different tumors and healthy tissues obtained from the TGCA database is shown in Figure 4. By analyzing clinical data, it was found that FEN1 protein expression is related to the aggressiveness of epithelial ovarian and breast cancer and the prognosis of patients, and high expression of FEN1 is positively correlated with the survival rate of patients [56]. Therefore, FEN1 may serve as a key biomarker for ovarian and breast cancer [57,58,59]. When FEN1 is abnormally mutated, the expression or function of FEN1 is altered, which may lead to the malignant transformation of normal cells or increase the susceptibility of patients to other carcinogens or the environment [60]. FEN1 gene polymorphisms may interact with gallstones and synergistically increase the risk of gallbladder cancer [60].

A mutant mouse model with FEN1 point mutation (F343A/F344A, FFAA) was established, which specifically eliminated FEN1-PCNA interactions [14]. Then, we found FFAA mutations lead to RNA primer removal and LP-BER defects, resulting in extensive DNA breaks. These breaks activated the G2/M checkpoint protein Chk1 and induced near-tetraploid aneuploidy, promoting the development of aneuploid cancers. Overexpression of FEN can promote the proliferation, cell cycle, migration or invasion, drug resistance, xenograft growth and epithelial mesenchymal transition (EMT) of hepatoma cells [61]. In addition, the abnormally high expression of IGF2BP2, a key gene in the m6A signaling pathway, was found in HCC cells. IGF2BP2 could directly recognize and bind to the m6A site of FEN1 mRNA, enhance the stability of FEN1 mRNA and promote the progression of liver cancer [62]. FEN1 inactivated the P53 signaling pathway by recruiting ubiquitin specific protease 7 (USP7) to prevent the ubiquitination and degradation of murine double minute 2 (MDM2). Further analysis of clinical samples also found that high expression of FEN1 was associated with aggressive tumor behavior and may directly play a role in the progression of carcinoma in situ to aggressive disease [63]. TGF-β 1 was found to promote epithelial mesenchymal transformation (EMT) phenotype through the up-regulation of FEN1 and down-regulation of MiR-140-5p, leading to the metastasis of HCC cells [64]. MiR-140-5p can directly down-regulate the expression of FEN1 in overexpressed HCC cells, thus inhibiting EMT.

FEN1 can not only promote the occurrence and development of solid tumors, but also has a certain role in hematological tumors. Through analysis of clinical data, FEN expression was found to be abnormally elevated in CML cells and bone marrow samples from patients [65]. FEN1 participates in the error-prone end-joining of the ALT-EJ repair mechanism, and the activation of ALT-EJ can inhibit c-NHEJ activity to a certain extent. When suppressing abnormally upregulated FEN1, it was found that it had a positive effect on CML in imatinib (IM) resistant cells. This was through inhibiting cell proliferation and promoting cell apoptosis, accompanied by the excessive and persistent accumulation of unrepaired DSB, thus achieving a good anticancer effect [65]. The presence of the FEN1 mutant rs174538 A allele may serve as a novel detectable and predictive biomarker for childhood acute lymphoblastic leukemia (ALL) [66]. In conclusion, the aberrant expression of FEN1 in a variety of tumors makes it an effective target and prognostic biomarker for cancer therapy.

### 3.2. Other Diseases

TNR expansion and deletion are closely related to human neurodegenerative diseases. FEN1 is also involved in the mechanism of triple repeat expansion in neurological diseases. FEN1 can interact with Pol β to mediate CAG repeat expansion. When DNA was damaged, FEN1 promoted CAG repeat deletion during repairing [67]. This effect led to the instability of TNR, which promotes the development of neurological diseases. In addition, FEN1 responded to apoptotic stimuli by breaking apoptotic DNA fragments through its 5′ exonuclease (EXO) and Gap endonuclease (GEN) activity [50]. Under the stimulation of pathological conditions, FEN1 will undergo various mutations. It was found that the EXO and GEN activity of FEN1 point mutations containing E160D were affected, resulting in the accumulation of damaged DNA fragments. Incomplete digestion of apoptotic DNA fragments may induce autoimmune diseases (systemic lupus erythematosus) and chronic inflammatory responses [68].

## 4. Development of FEN1 Inhibitors

### 4.1. Structural Design of Small-Molecule Inhibitors

Small-molecule inhibitors refer to a class of organic compounds with small molecular weight that can cross cell membranes by targeting the active site of a protein, reducing its activity or blocking the cellular response. In recent years, small-molecule inhibitors for various diseases have been developed accordingly. By analyzing the interactions between drugs and targets through biological information, drug design can be divided into de novo drug design, fragment-based drug design, and molecular docking [69]. De novo drug design mainly analyzes the direct interaction between the drug and the target, to design new molecules that match or complement the active site of the target. Cao et al. designed a small-molecule compound that binds to the active site of the human angiotensin-convert enzyme 2 (ACE2) by analyzing the interaction between the SARS-CoV-2 spike protein and ACE2 receptor. The novel coronavirus was effectively inhibited by blocking its binding to SARS-CoV-2 [69]. Molecular fragment-based drug design is a low molecular weight ligand (~150Da) that can recognize binding to biological macromolecules [70]. Molecular docking is a method used in drug discovery using computer algorithmic structures to predict drug–target interactions at the molecular level based on the chemical and three-dimensional structures of the target. The chemical structure and three-dimensional structure of the target of the drug action are analyzed, and the active site is analyzed by computer imaging. According to the interaction between these active sites and ligands, the drug molecules to be designed are obtained by virtual screening. Most of the inhibitors targeting FEN1 mentioned in this paper were obtained by the virtual screening of the FEN1 structure. The molecular docking models of the magnesium ions (Mg^2+^) sites of hFEN1 and the N-hydroxyurea compound are displayed in Figure 5, which are modified from the Protein Data Bank by PyMol (https://pymol.org/2/) (accessed on 24 May 2022) [24].

### 4.2. Small-Molecule Inhibitors Targeting FEN1

The DNA damage repair system is very important to maintain the survival and development of the body, and the abnormal expression or mutation of genes in this system also plays a major role in promoting the occurrence and development of tumors. Therefore, there has been an endless stream of research on targeted drugs for the development of genes and proteins related to the DNA damage repair system [71]. FEN1 is a metal ion-dependent and structure-specific nuclease that excises 5′ single chain flaps caused by lagging chain synthesis of DNA replication or chain replacement reactions in LP-BER [21]. Oligonucleotides cleaved by FEN1 produce a 5′ phosphorylated terminus suitable for ligating to ensure genomic stability [67]. A major feature of malignancies is the rapid and unlimited DNA replication that requires over activation of the DNA repair system to recover errors accumulated during the process [10]. In some cases, overexpression of FEN1 can enhance cancer susceptibility, accelerate malignancy progression, and reduce cancerous patient survival [72]. Therefore, FEN1 may be a druggable target, and small-molecule inhibitors targeting FEN1 have also emerged. Recently developed FEN1 inhibitors are listed in Table 1.

#### 4.2.1. FEN1 Inhibitors of N-hydroxy Urea

Dozens of inhibitors targeting FEN1 through high-throughput screening were filtered, which had more or less of an effect on the treatment of cancer [73]. FEN1 inhibitor-based four chemical structure N-hydroxy urea compounds (Compound #2, Compound #8, Compound #16 and Compound #20) had significant effects on several tumor models, according to previous reports [58,73]. In particular, Compound #8 significantly slowed tumor growth in mice inoculated with HCT116 and HCC1806 cells. FEN1 inhibitors may be effective drugs for the treatment of HR and other cancer deficiency in DNA repair and DNA damage checkpoint pathways. However, its low potency and pharmacokinetic profile may not be of clinical utility.

BRCA1 and BRCA2-deficient breast and ovarian cancers have HR deficiencies [59]. Although in the clinics, PARP inhibitors have been used to treat cancers with DSB repair defects, such as BRCA1 and BRCA2 defect cancers, the drug resistance to PARP inhibitors is now a major problem facing the clinic. Recent studies have found that the FEN1-specific inhibitor Compound #8 is abnormally sensitive to cancers with BRCA1 and BRCA2 deficiencies. In vitro studies have found that when Compound #8 was used to treat PEO4 versus PEO1 cells, the level of 53BP1 lesions in PEO1 cells was higher compared to PEO4 cells, consistent with the reduction in HR in the BRCA2-deficient PEO1 cells. Replication defects and DNA damage caused by FEN1 inhibition promote cell death by activating caspase [59]. At the same time, in vivo studies found that the tumor growth of BRCA deficiency in Compound #8 treatment was significantly reduced. These results suggest that the FEN1 inhibitor Compound #8 can reach the tumor site of mice and inhibit tumor growth in the manner observed after in vitro cell line treatment.

Compound #20 was used to target FEN1 as previously reported, which is an N-hydroxy urea derivative that specifically inhibits FEN1 activity and was the most potent FEN1 inhibitor tested in vitro at that time [74]. This therapy was selectively toxic to Pol β-deficient ovarian cancer cells, but not to XRCC1-deficient ovarian cancer cells. These results suggest that FEN1 inhibitors may be a possible therapeutic agent for the treatment of HR and other cancers with defective DNA repair and DNA damage pathways. Due to their inefficiencies and pharmacokinetic characteristics, the compounds tested here are unlikely to have clinical utility. However, its low potency and pharmacokinetic profile may not be of clinical utility.

#### 4.2.2. FEN1i

FEN1 overexpression is associated with an increased aggressive phenotype and poor prognosis of cancer after cisplatin chemotherapy, while inhibition of Pol β or XRCC1 can enhance the sensitivity of ovarian cancer to cisplatin treatment [16]. The abovementioned study found that FEN1 was synthetically lethal in Pol β deficiency but not synthetically lethal in XRCC1, ATM or MRE11 deficiencies. The authors screened the 391275 compounds by high-throughput screening (HTS) to synthesize a new series of FEN1 inhibitors (FEN1i). In vitro tests have found that FEN1 deletion or gene inactivation could reverse cisplatin tolerance in ovarian cancer cells, while FEN1 small-molecule inhibition increased platinum sensitivity. When FEN1i and cisplatin-resistant cells were combined with cisplatin, there was an increase in nuclear γH2AX nuclear foci, G2/M cell cycle arrest and an increase in apoptotic cells. In the course of PEO4 cell therapy, it was also found that the combination of FEN1/cisplatin also increased cell death compared with cisplatin or FEN1 monotherapy. In addition, FEN1 also physically interacted with Importin β [16]. After cisplatin therapy, the FEN1 protein increased and metastasized to the nucleus, and the nuclear localization of FEN1 was mediated by Importin β pathway. Using an introduction Importin β inhibitor can block the shift of FEN1 to the nucleus and resensitize A2780cis cells to cisplatin therapy. However, the mechanism of the interaction of FEN1 with the introduction of protein β needs further study.

#### 4.2.3. FEN1i #2

Estrogen receptor α (ERα), a transcriptional regulator, has been found to be abnormally expressed in breast cancer. ERα-positive patients often develop drug resistance when treated with tamoxifen, which makes treatment difficult [75]. FEN1 can enhance the transcriptional activity of ERα by facilitating the recruitment of ERα transcription complex co-activators. Inhibition of FEN1 can activate the proteasome-mediated degradation of ERα, leading to a decrease in ERα expression and inhibiting the proliferation of tumor cells. A novel specific inhibitor targeting FEN1 from 465,195 compounds through high-throughput compound screening was identified, named as FEN1i #2 [76]. FEN1i #2 regulated ERα activity by recruiting BRG1 through the formation of γH2AX. FEN1i #2 was demonstrated to have a specific blocking effect on ERα-positive breast cancer. The sensitivity of tamoxifen-resistant cell lines to the FEN1 blockade was also increased and may be used to treat advanced breast cancer in the future, providing a new targeted therapy for tamoxifen-resistant cases. FEN1 is one of the important factors for poor prognosis in ERα-positive patients. Previous development of FEN1 inhibitors has been primarily associated with chemical sensitization or as part of synthetic lethal interactions, but it is not an effective therapeutic strategy in itself [20,76]. The authors screened out ERα-specific targeted inhibitors by compounds and found almost all of them ineffective in the analysis of 150 additional proteins, but there may still be some non-targeted effects that require further improvements to the chemicals to further clinically develop FEN1-targeted therapies for breast cancer.

#### 4.2.4. SC13

FEN1 is overexpressed in HeLa cells, particularly after IR irradiation. SC13, a specific inhibitor targeting FEN1 induced impaired DNA damage repair mechanisms that leads to cancer cell apoptosis, thereby enhancing the IR sensitivity of cervical cancer [77]. At the same time, paclitaxel in combination with the FEN1 inhibitor SC13 can significantly improve the efficacy, indicating that there is a synergistic mechanism between the two compounds [78]. SC13 affect Okazaki fragment maturation and LP-BER by inhibiting FEN1 in vitro. In cells, SC13 inhibits cancer cell proliferation and induces chromosomal instability and cytotoxicity, making cancer cells more sensitive to DNA damage reagents. Importantly, in mouse models, SC13 sensitized the tumor to the chemotherapy drug, reduced the dose of the chemotherapy drug and its toxic side effects, and hindered the progress of tumor development. The combination treatment of SC13 and paclitaxel significantly inhibited the expression of CDK2/4 and cyclins, thereby significantly inducing cell cycle arrest in cervical cancer cells. It was shown that SC13 combination therapy sensitized cancer cells to low-concentration paclitaxel treatment, which meant that the side effects of paclitaxel could be avoided, and targeting FEN1 may be a new strategy for tumor-targeted therapy of cervical cancer. When SC13 was combined with low-dose camptothecin, the killing effect on tumors was significantly enhanced. The results demonstrated that the two have a synergistic effect when used together, and increased the killing effect on tumor cells through the internal mitochondrial apoptosis pathway [79].

#### 4.2.5. Other Inhibitors

Flavonoids are defined as a group of natural substances, which are widely distributed in fruits, vegetables, various plants, etc., and have strong medicinal value. Myricetin (3,3′,4′,5,5′,7-Hexahydroxyflavone cannabiscetin), a class of flavonoids, acts as a FEN1 inhibitor and was sensitive to FEN1 overexpression in colorectal cancer [80]. Most of the FEN1 inhibitors are small organic molecules known to have serious side effects on the body′s metabolism, and may also cause drug resistance due to genetic reasons, and mutations occur in the body. Therefore, a novel substrate-competitive FEN1 inhibitor was designed, which exhibited a high inhibitory efficiency against FEN1 for substrates containing phosphorothioate bonds at the FEN1 cleavage site of DNA substrates [81]. This inhibitor was modified by phosphorothioate to slow down FEN1 catalysis and inhibit tumor growth.

### 4.3. Defects

Although progress has been made in the development of small-molecule inhibitors, its defects cannot be ignored. Small molecules have simple structures and are relatively easy to synthesize. As research progresses, we have a vast library of molecules, and more sophisticated and efficient screening techniques, and it is more difficult to discover entirely new molecular entities [82]. In addition, the development of small-molecule inhibitors is based on a single principle, and only inhibits the function of target proteins through site-occupying binding [83]. Small molecule-targeted drugs also have non-negligible side effects, such as skin and mucous membrane damage, resulting in rashes, ulcers, diarrhea, and severe liver damage [77]. Furthermore, small-molecule inhibitors can also develop resistance, which is mainly due to genetic mutations [83]. The FEN1 inhibitor developed in this paper has limitations in pharmacokinetics and efficacy, and the effect has only been demonstrated in animals, with the possibility that it could not be reproduced in humans and not be used for clinical treatment. The small-molecule inhibitors developed targeting FEN1, as discussed in this paper, and despite their effectiveness at the cellular and animal levels, are still a long way from clinical studies.

## 5. Conclusions

In this review, we summarize the development and application strategies of small-molecule inhibitors targeting FEN1. These small molecule inhibitors could be used alone as monotherapy or combined with other means of cancer therapy to reduce the resistance of cancer cells [84]. These small molecule inhibitors affect Okazaki fragment maturation, genomic stability or DNA damage repair pathways (BER, HRR) by inhibiting FEN1, then killing tumor cells or increasing tumor sensitivity to chemotherapy drugs [16,59,78].

FEN1 has been found to have abnormal expression or mutation in a variety of solid tumors, which has promoted the proliferation and migration of tumor cells [50]. FEN1 has also been found to be closely related to tumor chemotherapy resistance, and is considered to be a marker of metastasis and poor prognosis of various malignant tumors [61]. Therefore, FEN1 is considered to be an effective target for the treatment of tumors. FEN1 inhibitors based on N-hydroxy structure have shown a powerful effect on tumors with defective HR [73,74]. Activation of FEN1 in tamoxifamine-resistant breast cancer regulates ERα activity by stabilizing chromatin interactions. FENi #2 was used to treat breast cancer by targeting FEN1 and disrupting chromatin stability [76]. The FEN1 inhibitor SC13 developed in our laboratory affected the sensitivity of Okazaki fragment maturation and LP-BER pathway in vitro enhanced tumors to chemotherapeutic agents [78]. Thus, small-molecule inhibitors targeting FEN1 enhanced the therapeutic effect of tumors by affecting the process of DNA replication and DNA damage repair, leading to cell cycle disorder, cell necrosis or apoptosis. The development of small-molecule inhibitors targeting FEN1 is a good strategy for treating cancers.

It is still necessary to use computer technology and bioinformatics and other related technologies to search for the small-molecule inhibitors or discover new structural compounds targeting FEN1.The structure of the agent should be optimized so that it can achieve the effect of high efficiency and low toxicity, which can be further applied to clinical treatment. Combination therapy with inhibitors and other treatments is also a trend for the treatment of cancers in the future. We look forward to new insights and applications on FEN1 inhibitors in cancer therapy.

## Figures and Tables

**Figure 1 biomolecules-12-01007-f001:**
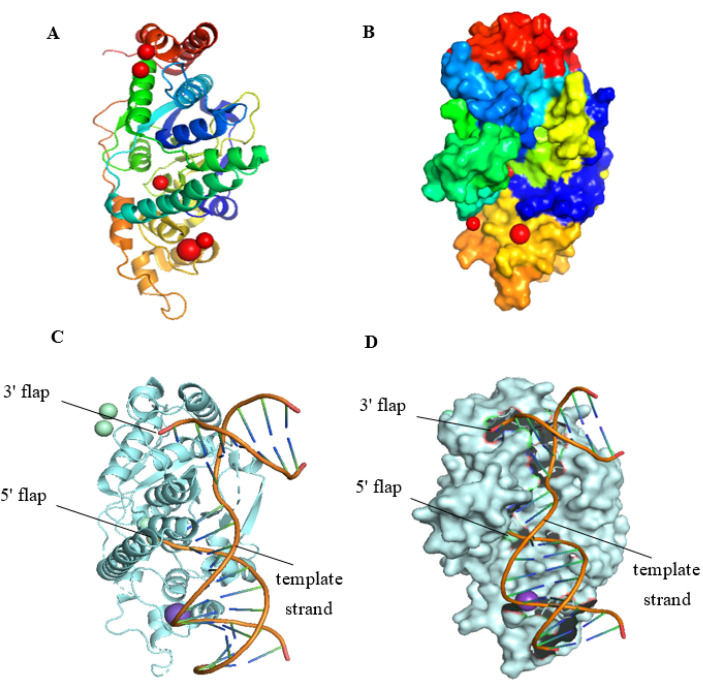
The structure of the hFEN1. (**A**) The published hFEN1 secondary structure (PDB ID: 5UM9). (**B**) The published hFEN1 three-dimensional structure (PDB ID: 5UM9). (**C**) The published hFEN1-DNA secondary structure (PDB ID: 3Q8L). (**D**) Superposition of double-flap DNA in the hFEN1 pre-threading structure (colored) and the published threading structure (white, PDB ID: 3Q8L). The template strand, 5′flap and 3′flap are labeled.

**Figure 2 biomolecules-12-01007-f002:**
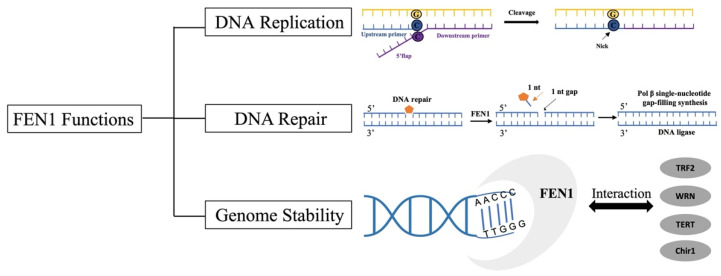
Major functions of FEN1.

**Figure 3 biomolecules-12-01007-f003:**
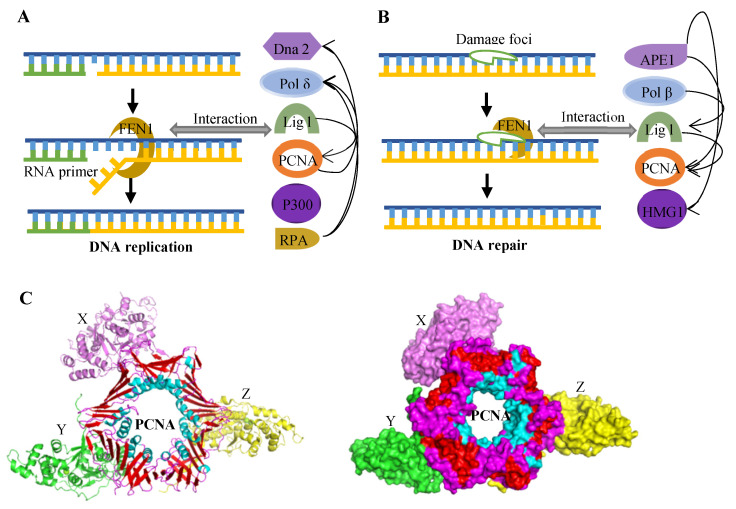
The proteins interact with FEN1. (**A**) Proteins act in concert with FEN1 during DNA replication. (**B**) Proteins act in concert with FEN1 during DNA repair. Arrows indicate stimulation of enzyme activity. (**C**) The structure of the human FEN1-PCNA complex (PDB ID: 1UL1). Three FEN1 molecules are colored in pink (X), green (Y) and yellow (Z), and three sections of PCNA are respectively colored as red, blue and violet.

**Figure 4 biomolecules-12-01007-f004:**
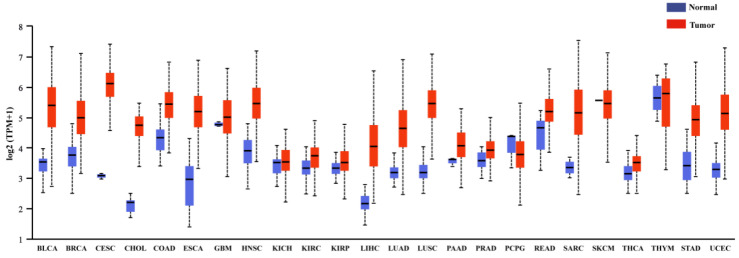
Expression of FEN1 across TCGA cancers (with tumor and normal samples).

**Figure 5 biomolecules-12-01007-f005:**
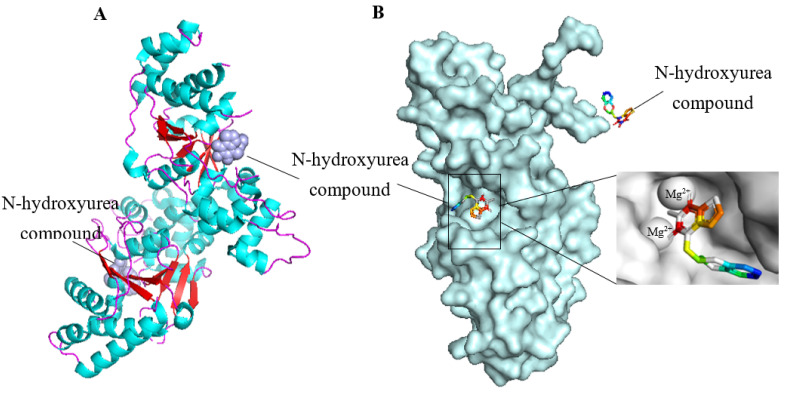
The molecular docking models of hFEN1 and small molecule compounds. (**A**) Superposition of N-hydroxyurea compound (purple) in the secondary structure hFEN1 (PDB ID: 5FV7). (**B**) The structure of N-hydroxyurea compound (colored) in the hFEN threading structure (white, PDB ID: 5FV7). The N-hydroxyurea compound and Mg^2+^ are labeled.

**Table 1 biomolecules-12-01007-t001:** Recently developed FEN1 inhibitors.

Name	Chemical Structure	Reference
Compound #2	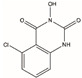	[73]
Compound #8	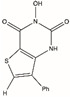	[59,73]
Compound #16	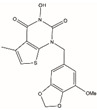	[73]
Compound #20	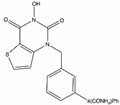	[73,74]
FEN1i	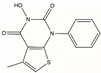	[16]
FEN1i #2	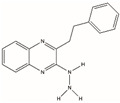	[75,76]
SC13	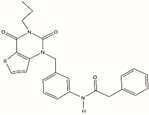	[77,78,79]
Myricetin	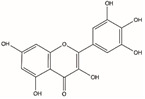	[80]
Novel FEN1 inhibitor	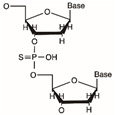	[81]

## Data Availability

Not applicable.

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
