# Peer review of "Small-Molecule Inhibitors Targeting FEN1 for Cancer Therapy"

_biomolecules, 2022, doi:10.3390/biom12071007_

Round 1

Reviewer 1 Report

The manuscript “Small molecular inhibitors targeting FEN1 for cancer therapy” by Yank and colleagues provides a review of the structure, function protein interactions and roles in human disease of FEN1along with the relatively short presentation of the existing inhibitors that have been identified.  While the initial section up to the presentation of the inhibitors is interesting, however, the authors do not necessarily shed new light on any specific aspect of Fen 1 and more comprehensive and thorough review of fen 1 are available.  The discussion of the various inhibitors represents a relatively small portion of the manuscript and is underdeveloped.  A more thorough analysis of the individual inhibitors would greatly enhance the impact of this paper and the interest around the target.  A discussion of the chemistry, how specific interactions are similar or different with FEN1 would be welcome.  The specific differential effects on FEN1 biochemical activity and in specific pathways including Okazaki fragment processing.  The specific impact on cellular activity and modulation of the FEN1 dependent pathways would enhance the presentation.  Reporting the cellular effect is an important start but expansion to provide more insight into these important pathways and how then may regulation the more complex cellular process is warranted.  In addition, in vivo activity needs to be addressed.  If specific inhibiotrs have not been tested in vivo, a discussion on why that is or what would be expected is in order.  If the inhibitors are simply tool compounds and not necessarily being developed as therapeutics a discussion of the limitations or reasons would be welcome.  Overall the topic is interesting but a more in depth analysis of the literature and compounds is needed to ensure the review is relevant and comprehensive.           

Reviewer 2 Report

The authors of the review "Small molecule inhibitors targeting FEN1 for cancer therapy" attempt to summarize how FEN1 is a viable target for cancer treatment.

This manuscript is potentially interesting and would be helpful to the scientific community.

Yet, overall enthusiasm for the manuscript is lowered by the following issues:

1: Help from a professional editor would benefit this manuscript. In several occasions the sentence structure does not make sense, making the manuscript difficult to read.

2: The authors make a point to highlight a role of FEN1 in multiple different DNA repair pathways, yet only make an effort to describe FEN1 function in BER repair. Sections should be added to describe function in the other repair pathways.

3: The authors also do not mention that FEN1 plays a role in telomere stability.

4: The conclusion is confusing. It appears to be focused on CART therapies with a small section on FEN1. Then there is no attempt to connect FEN1 to CART therapy. The conclusion would be better of discussing how inactivation of FEN1 would affect different DNA repair pathways and how this would enhance tumor therapy.

Round 2

Reviewer 1 Report

All issues have been adequately addressed

Author Response

We appreciate the positive comment of this reviewer to our manuscript. We also  thank this reviewer for his/her help in improving our manuscript.

Reviewer 2 Report

The revision of the manuscript is much improved. The authors were extremely responsive to previous comments. There are no concerns with current version of manuscript.

Author Response

(The authors gave the same response as above.)
